# Brief communication: A momentum-conserving superposition method applied to the super-Gaussian wind turbine wake model

Frédéric Blondel[1]

[1]IFP Energies nouvelles, 1-4 Av. du Bois Préau, 92852 Rueil-Malmaison

**Correspondence:** Frédéric Blondel (frederic.blondel@ifpen.fr)

**Abstract.** Accurate wind farm flow predictions based on analytical wake models are crucial for wind farm design and layout optimisation. In this regard, wake superposition methods play a key role and remain a substantial source of uncertainty. Recently, new models based on mass and momentum conservation have been proposed in the literature. In the present work, such methods are extended to the superposition of super-Gaussian type velocity deficit models, allowing the full wake velocity deficit estimation and design of closely packed wind farms.

## 1 Introduction

Wind farm design and layout optimisation rely on analytical flow models due to a large number of configurations to be evaluated and the computational efficiency of such numerical methods. A typical wind farm flow solver consists of a combination of several sub-models, including *a minima* a velocity deficit model, a wake-added-turbulence (WAT) model, and possibly a wake deflection model, a blockage model, and a coupled wake/atmospheric-boundary-layer model. The velocity deficit and WAT models usually apply to a single wind turbine: wake superposition methods accumulate the wakes and estimate a wind farm power production for given environmental conditions. Concerning the superposition of velocity deficits, the available methods lacked theoretical justification, until the recent work of Zong and Porté-Agel (2020) and Bastankhah et al. (2021). In these studies, analytical solutions for the velocity deficit superposition are proposed based on the mass and momentum conservation principle. These superposition methods assume Gaussian-shaped velocity deficit profiles. In the present article, the approach of Bastankhah et al. (2021) is extended to super-Gaussian wake velocity deficit profiles. Such models, proposed in Shapiro et al. (2019) and later refined in Blondel and Cathelain (2020), allow for the evaluation of the velocity deficit over the full wake. On the contrary, the Gaussian-based approaches are limited to the far-wake. Apart from preventing the appearance of unrepresentable numbers, this allows the study of closely packed wind farm layouts. Indeed, some offshore wind farms such as Lillgrund exhibit small wind turbine inter-distances, down to 3.3 wind turbine diameters. Considering such super-Gaussian velocity profiles together with the Bastankhah et al. (2021) superposition method, an integral has no analytical solution, and an approximation is proposed and compared with the numerical solution. It is also shown in Section 3 that the method proposed in Bay et al. (2022) leads to similar results in terms of centerline velocity deficit and is suited for wind-farm power predictions. The new superposition method has more robust theoretical foundations than the traditionally used local-linear-sum (LLS)

 superposition technique (method C in Zong and Porté-Agel (2020)), and its applicability is demonstrated based on the large Horns-Rev wind farm.

## 2 Extension of the Bastankhah et al. (2021) model

### 2.1 Model derivation

In Bastankhah et al. (2021), the conservation of momentum deficit for multiple wakes takes the form:

$$30 \quad \int_{\tilde{A}} \left( u_0 c_n f_n - (c_n f_n)^2 - 2 c_n f_n \sum_{i=1}^{n-1} c_i f_i \right) d\tilde{A} \approx \frac{\tilde{T}_n}{\rho}, \tag{1}$$

with $c_n$ the maximum velocity deficit of turbine $n$, $i$ the index of the turbines upwind of turbine $n$, $f_n$ the self-similar function, $\tilde{A} = \pi \tilde{r}^2$ the rotor surface with $\tilde{r} = r/d_0$ and $d_0$ the wind turbine diameter, $\tilde{T}_n$ the thrust force of the unit diameter rotor, $u_0$ the undisturbed wind velocity, and $\rho$ the fluid density. Based on comparisons to numerical results from a large-eddy simulation (LES) solver, a modified form was proposed in Bastankhah et al. (2021): the factor two in the left-hand side of Eq. (1) is

35 dropped.

Let us consider the original form, Eq. (1). Given a super-Gaussian shape function $f_n$, a solution for $c_n$ is sought. Following Blondel and Cathelain (2020), the shape function reads $f_i = \exp\left( -\tilde{r}_i^k / 2\tilde{\sigma}_i^2 \right)$, with $k = k(\tilde{x})$ the super-Gaussian order and $i$ or $n$ the index of a wind turbine. In the following, we assume that the turbines are sorted from the most upwind to the most downwind, and for two turbines $i$ and $n$, we have $i < n$.

Here, as indicated by the tilde, the radius and the super-Gaussian characteristic width are both normalized by the wind turbine diameter $d_0$, such as: $\tilde{r}_i = \sqrt{(y-y_i)^2 + (z-z_i)^2}/d_0$ and $\tilde{\sigma}_i = \sigma_i/d_0$. The following integrals are defined in terms of the gamma function $\Gamma$:

$$\int_{\tilde{A}} f_n d\tilde{A} = \frac{\pi}{k} \Gamma\left(\frac{2}{k}\right) 2^{2/k+1} \tilde{\sigma}_n^{4/k}, \qquad \int_{\tilde{A}} f_n^2 d\tilde{A} = \frac{2\pi}{k} \Gamma\left(\frac{2}{k}\right) \tilde{\sigma}_n^{4/k}, \quad \text{and} \quad \int_{\tilde{A}} f_n f_i d\tilde{A} = \mathcal{I}. \tag{2}$$

No analytical solution could be found for the last integral, denoted $\mathcal{I}$. Inserting Eqs. (2) into Eq. (1) leads to:

$$45 \quad u_0 c_n \frac{\pi}{k} \Gamma\left(\frac{2}{k}\right) 2^{2/k+1} \tilde{\sigma}_n^{4/k} - c_n^2 \frac{2\pi}{k} \Gamma\left(\frac{2}{k}\right) \tilde{\sigma}_n^{4/k} - 2 c_n \sum_i^{n-1} c_i \mathcal{I} \approx \frac{\tilde{T}_n}{\rho}. \tag{3}$$

Using the thrust coefficient $C_{T_n} = 8\tilde{T}_n / \left( \pi \rho \tilde{d}_0^2 <u_{n-1}>_{(n,x_n)}^2 \right)$, the operator $<>_{(n,x_n)}$ denoting the spatial averaging over the frontal projected area of rotor $n$ at $x = x_n$, and $u$ the streamwise velocity component as in Bastankhah et al. (2021), one obtains:

$$c_n^2 - c_n 2^{2/k} \left( u_0 - 2 \sum_i^{n-1} \frac{c_i}{2^{2/k}} \frac{k\mathcal{I}}{2\pi \Gamma\left(\frac{2}{k}\right) \tilde{\sigma}_n^{4/k}} \right) + \frac{k C_{T_n}}{16} \frac{<u_{n-1}>_{(n,x_n)}^2}{\Gamma\left(\frac{2}{k}\right) \tilde{\sigma}_n^{4/k}} \approx 0. \tag{4}$$

Let us introduce a modified integral $\mathcal{J} = k\mathcal{I} / \left( 2^{2/k} \pi \Gamma \left( \frac{2}{k} \right) \tilde{\sigma}_n^{4/k} \right)$. After straightforward manipulations, and assuming $u_0 = u_h$, i.e., a constant, shear-free inflow, the solution for $c_n$ reads:

$$\frac{c_n}{u_h} = \left( 1 - \sum_{i=1}^{n-1} \frac{c_i}{u_h} \mathcal{J} \right) \left( 2^{2/k-1} - \sqrt{ 2^{4/k-2} - \frac{k C_{T_n} \left( \frac{<u_{n-1}>_{(n,x_n)}}{u_h} \right)^2}{16 \Gamma \left( \frac{2}{k} \right) \tilde{\sigma}_n^{4/k} \left( 1 - \sum_{i=1}^{n-1} \frac{c_i}{u_h} \mathcal{J} \right)^2} } \right). \tag{5}$$

The modified form is obtained by using a modified $\mathcal{J}$ together with Eq. (5) and $\mathcal{I}^{mod} = \mathcal{I}/2$:

$$\mathcal{J}^{mod} = \frac{k \mathcal{I}^{mod}}{2^{2/k} \pi \Gamma \left( \frac{2}{k} \right) \tilde{\sigma}_n^{4/k}}. \tag{6}$$

## 55    2.2    Approximate solutions of the integral $\mathcal{I}$

In a first approach, hereafter referred to as the $Gauss$ approach, one may assume a Gaussian behaviour of the model to evaluate $\mathcal{J}$, as done in Bay et al. (2022). One obtains, see Bastankhah et al. (2021):

$$\mathcal{J}_{Gauss}^{mod} = \frac{\pi \tilde{\sigma}_i^2 \tilde{\sigma}_n^2}{\tilde{\sigma}_i^2 + \tilde{\sigma}_n^2} \exp \left( -\frac{(\tilde{y}_n - \tilde{y}_i)^2}{2(\tilde{\sigma}_n^2 + \tilde{\sigma}_i^2)} \right) \exp \left( -\frac{(\tilde{z}_n - \tilde{z}_i)^2}{2(\tilde{\sigma}_n^2 + \tilde{\sigma}_i^2)} \right). \tag{7}$$

Alternatively, in a second approach hereafter referred to as the $kEquiv$ approach, one may first consider aligned turbines
$(\tilde{y}_i - \tilde{y}_n = 0, \tilde{z}_i - \tilde{z}_n = 0)$ and later correct the integral for the lateral distance between the rotors using a function $\delta(\tilde{y}, \tilde{z})$. This function is identified from the Gaussian solution. A second approximation consists in considering an equivalent super-Gaussian order, $k_{eq} = 1/2 \, (k_i + k_n)$. Under these hypotheses, the integral $\mathcal{I}$ takes the form:

$$\mathcal{I}_{kEquiv}^{mod} = \frac{\pi \Gamma \left( 2/k_{eq} \right) 2^{2/k_{eq}+1} \tilde{\sigma}_i^{4/k_{eq}} \tilde{\sigma}_n^{4/k_{eq}}}{k_{eq} \left( \tilde{\sigma}_i^2 + \tilde{\sigma}_n^2 \right)^{2/k_{eq}}} \delta(\tilde{y}, \tilde{z}), \quad \text{with} \quad \delta(\tilde{y}, \tilde{z}) = \exp \left( -\frac{(\tilde{y}_n - \tilde{y}_i)^{k_{eq}}}{2(\tilde{\sigma}_n^2 + \tilde{\sigma}_i^2)} \right) \exp \left( -\frac{(\tilde{z}_n - \tilde{z}_i)^{k_{eq}}}{2(\tilde{\sigma}_n^2 + \tilde{\sigma}_i^2)} \right), \tag{8}$$

and Eq. (6) is used to calculate $\mathcal{J}_{kEquiv}^{mod}$. Another straightforward approach consists in tabulating the integral values (exclud-
ing the $\delta(\tilde{y}, \tilde{z})$ function) and linearly interpolating between the data, which is the one retained in practice. For a quantitative comparison, the proposed analytical approximations of the integral $\mathcal{J}$ are compared to the numerical integration. An interval of $0.2 \le \tilde{\sigma}_i, \tilde{\sigma}_n \le 2.5$ is considered for the characteristic width, and several intervals $2 \le k_i, k_n < max_k$ are considered for the super-Gaussian order, with $2 < max_k \le 8$. The bounding values are representative of the very near wake of a wind turbine under laminar flow conditions and the very far wake ($\tilde{x} > 15d_0$) under highly turbulent conditions: the typical operating
range of a turbine in a wind farm is covered. Among the characteristic width and super-Gaussian order intervals, 15 values are sampled. Regarding the maximum super-Gaussian order, six equally-spaced values are sampled. For each set of four inputs, and for a given maximum super-Gaussian order, the analytical approximations are evaluated, and the error is computed ($error = (|\mathcal{J}_{Analytical}| - |\mathcal{J}_{Numerical}|) / |\mathcal{J}_{Numerical}|$). The numerical evaluation is based on the scipy (Jones et al. (2001)) "integrate.quad" integration routine, and extends from 0 to $6 \cdot max(\sigma_i, \sigma_n)$. Then, for each $max_k$, the average and maximal er-
ror are computed and reported in Figure 1. From these results, the so-called $kEquiv$ method seemingly outperforms the $Gauss$ method and should be preferred. However, it will be shown in Section 3 that the impact on the velocity deficit is limited.

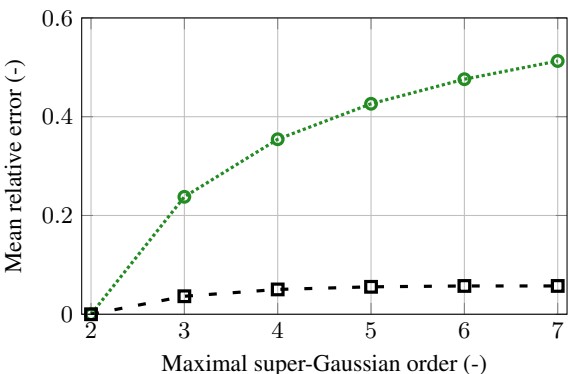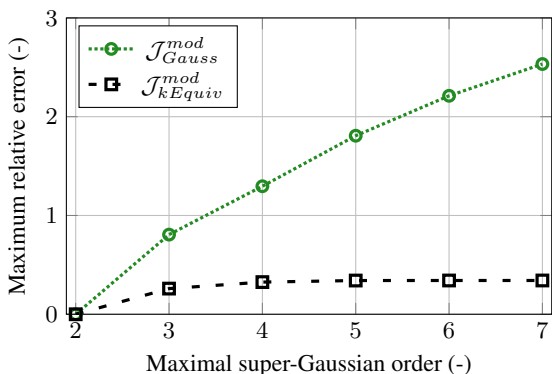

**Figure 1.** Mean (left) and maximal (right) relative error of the analytical integrals compared with the numerical evaluation of $\mathcal{J}$ as a function of the maximal considered super-Gaussian order

## 3 Results

In a recent study, Lanzilao and Meyers (2022) showed that the super-Gaussian model performed poorly compared with other models: for both Horns-Rev and the London Array wind farms, the predicted power production is far below the measured power from SCADA data. Due to these observations, the model is re-calibrated for the present study. The calibration procedure and the notations used hereafter follow the work of Cathelain et al. (2020). The main difference here lies in the use of a Gaussian profile in the far wake, i.e., $\lim_{\tilde{x}\to\infty} k(\tilde{x}) = 2$. The wake characteristic width is assumed to evolve linearly with axial distance:

$$\tilde{\sigma} = (a_s TI + b_s)\tilde{x} + c_s\sqrt{\left(\frac{1}{2}\frac{1+\sqrt{1-C_T}}{\sqrt{1-C_T}}\right)}. \tag{9}$$

The three parameters, $a_s$, $b_s$ and $c_s$, are used for both the super-Gaussian and the Gaussian model. The super-Gaussian order follows an exponential decay function:

$$k = a_f e^{b_f \tilde{x}} + c_f. \tag{10}$$

A Gaussian profile is assumed in the far wake, thus $c_f = 2$. The parameter $b_f$ controls the decay of the super-Gaussian order, and is taken as a function of the turbulence intensity. $a_f$ is chosen in such a way that the model fulfils the actuator-disk theory (see Cathelain et al. (2020)). This can be enforced numerically using a Newton fixed-point algorithm. To facilitate the implementation, this inversion is performed in a pre-processing stage, and a third-order polynomial fit is proposed:

$$a_f = -8.2635C_T^3 + 8.5939C_T^2 - 8.9691C_T + 10.7286. \tag{11}$$

The proposed calibration is not meant to be universal but dedicated to the present study. Future work will be dedicated to a calibration that is reliable in both near and far-wake regions. Table 1 provides the list of the model coefficients used in this study, obtained using a differential evolution algorithm and a set of nine LES simulations.

**Table 1.** Coefficients of the super-Gaussian wake model

| $a_s$ | $b_s$ | $c_s$ | $a_f$ | $b_f$ | $c_f$ |
|-------|-------|-------|-------|-------|-------|
| 0.28 | 0.01 | $0.1 \times C_T + 0.1$ | Eq. 11 | $1.68 \exp\left(-25.98 TI\right) - 1.06$ | 2. |

## 3.1      Comparison against large-eddy simulations from Bastankhah et al. (2021)

For the model comparison, the numerical setup based on the aligned wind farm introduced in Bastankhah et al. (2021) is reproduced. A schematic view of the wind farm is given in Figure 2. It consist of five columns of three wind turbines, which are all included in the simulation. The velocity deficit studied later is extracted over a line passing through the hub of the wind turbines of the central line, as shown in the schema. The first column in our simulations is located at $\tilde{x} = 0$.

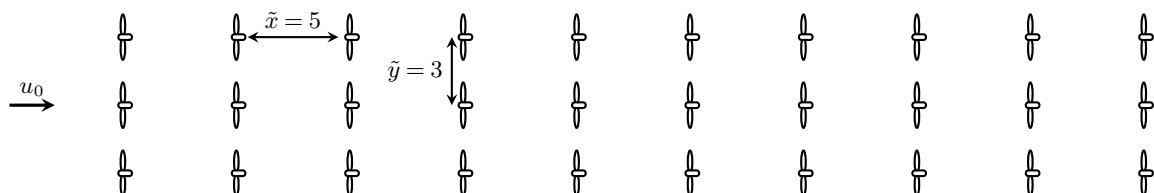

**Figure 2.** Schematic view of the simulated wind farm

The wind farm flow model builds upon the super-Gaussian model as described in Blondel and Cathelain (2020), using the calibration introduced in Section 3. The WAT model proposed in Ishihara and Qian (2018) is employed, together with a so-called "maximum-value" WAT superposition, see Niayifar and Porté-Agel (2016). A correction factor of 1.25 is applied on the maximum of added turbulence to match the results presented in Bastankhah et al. (2021). Following a convergence study, the rotor disks are discretized based on $12 \times 12$ polar grids. Velocity deficit and WAT due to upwind rotor wakes are evaluated at every point on the disk. Then, mean velocity and turbulence intensity are computed and used as an input for the wake models and rotor performance evaluation, i.e., the power and thrust coefficients are given as a function of wind speed. Using a polar discretization, the mesh cells are not uniform in size: the ones located near the edge of the disk are significantly larger than the ones near the hub. Thus, when computing the mean quantities, we use a weighted average whose ponderation is based on the mesh cell surface. In practice, in the case of aligned rotors, this tends to lower the wake effect since the higher velocity deficit is located at the rotor centre where the mesh cell's relative areas are the smallest. A blockage correction based on the vortex

cylinder flow model is used, see Branlard and Meyer Forsting (2020). The LLS method is compared to the present method, denoted MC (Momentum Conserving), with the two approximations for $\mathcal{J}^{mod}$, as well as a direct numerical evaluation of the integral, denoted $\mathcal{J}^{mod}_{Num}$.

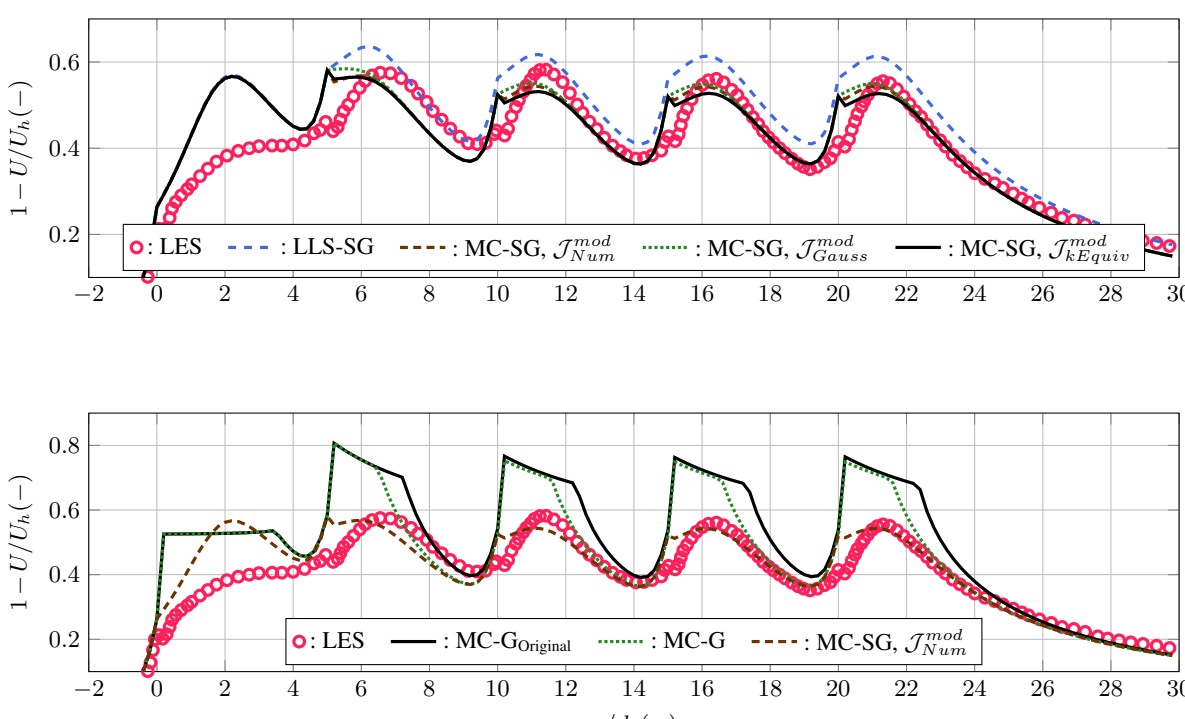

**Figure 3.** Centerline velocity deficit in the middle column of the farm, LES data scanned from Bastankhah et al. (2021). Top figure: comparison between LES, the LLS method together with the super-Gaussian model (LLS-SG), and the momentum-conserving method together with the super-Gaussian formulations (MC-SG). Bottom figure: comparison between LES and the momentum-conserving method and the Gaussian model, using the Original (MC-G$_{\text{Original}}$) and modified (MC-G) formulations, and a super-Gaussian formulation (MC-SG).

Figure 3 (top) shows that, compared with the LLS superposition method, the MC model predicts a lower velocity deficit in
both near and far wake regions which is more consistent with LES data. Moreover, the proposed analytical approximations of the integral $\mathcal{J}^{mod}$ are very close to the numerical approximation in the presented test case. At the rotor planes, discontinuities are observed. This can be partially attributed to the use of the modified momentum-conservation method which improves the results in the far wake as shown in Figure 3 (bottom) but does not fully respect the conservation laws, as detailed in Bastankhah et al. (2021). Using the un-modified formulation leads to very high near-wake velocity deficits or even unrepresentable numbers
in the presented test case. More than three diameters behind the wind turbine, the results based on the $\mathcal{J}^{mod}_{kEquiv}$ and $\mathcal{J}^{mod}_{Gauss}$ approximations are superimposed since the super-Gaussian order is close to 2. These observations validate the approach em-

ployed in Bay et al. (2022), despite the higher errors noticed in Figure 1. In practice, using a tabulated version of the integral is a fast and convenient approach. However, it does not circumvent the approximation based on the rotor distance function, $\delta(\tilde{y}, \tilde{z})$, since tabulating the complete integral results in large data files that are time-consuming to load. The global agreement against the LES dataset is satisfying. In the first turbine wake, the hub effect prevents a proper analysis of the results. For the second turbine, a good agreement is obtained with the LLS method, while the MC method underpredicts the velocity deficit. This behaviour, as noted in Bastankhah et al. (2021), is a consequence of the application of the modified momentum conservation law. For the following three turbines, a good agreement is obtained.

**Table 2.** Root-Mean-Square-Error of the analytical models against the LES results from Bastankhah et al. (2021)

| MC-G$_{\text{Original}}$ | MC-G | LLS-SG | MC-SG, $\mathcal{J}_{Gauss}^{mod}$ | MC-SG, $\mathcal{J}_{kEquiv}^{mod}$ | MC-SG, $\mathcal{J}_{Num}^{mod}$ |
|---|---|---|---|---|---|
| 0.160 | 0.145 | 0.078 | 0.059 | 0.055 | 0.056 |

For a more quantitative analysis, the Root-Mean-Square-Error (RMSE) between the different analytical models from $\tilde{x} = 0$ to $\tilde{x} = 30$ and the LES results are given in Table 2. First, the use of Gaussian wake models leads to a rather high error, due to the inaccuracy in the near-wake. This behaviour is expected, and we are here using the model outside of its definition domain, i.e., the Gaussian model is a far-wake model. Using the super-Gaussian model, the RMSEs fall below $8\%$. Whatever the approximation performed on the $\mathcal{J}$ integral, the momentum-conserving approach outperforms the LLS method: the RMSEs fall again from approximately 8 to less than $6\%$. Using the $\mathcal{J}_{Gauss}^{mod}$ approximation, the error is slightly higher compared with $\mathcal{J}_{kEquiv}^{mod}$ and $\mathcal{J}_{Num}^{mod}$. One should thus prefer one of these two formulations over the so-called $\mathcal{J}_{Gauss}^{mod}$ approximation.

### 3.2 Comparison against large-eddy simulations of the Horns-Rev wind farm from Porté-Agel et al. (2013)

The model predictions are also compared with large-eddy simulations of the Horns-Rev wind farm, as presented in Porté-Agel et al. (2013). With a minimal inter-turbine distance of seven diameters, this wind farm can not be considered as closely-packed. However, the availability of a large set of large-eddy simulation results makes it a good candidate for validation purposes. The inflow conditions are based on inflow velocity and turbulence intensity profiles scanned from Porté-Agel et al. (2013). Figure 4 compares the wind farm efficiency $\eta$ (predicted power divided by theoretical power without wake effect) over a wide range of wind directions $\theta$. We use the LES as a reference to avoid the uncertainties of SCADA measurements, mainly due to the wind direction changes during the $10\,\mathrm{min}$ averaging in the available data. The agreement between the analytical model and the LES dataset is overall good. Differences between the momentum-conserving superposition method and the LLS approach are noticed for wind directions where the wake effects are strong, typically at $\theta \approx \{222^o, 270^o, 312^o\}$. Around such directions, the lower velocity deficits predicted by the MC approach lead to lower wake losses and better efficiency of the wind farm which is more consistent with the LES data. Both the Gaussian and the super-Gaussian models predict the same wind farm efficiency whatever the wind direction: this is due to the large inter-turbine distances in the Horns-Rev wind farm. It confirms

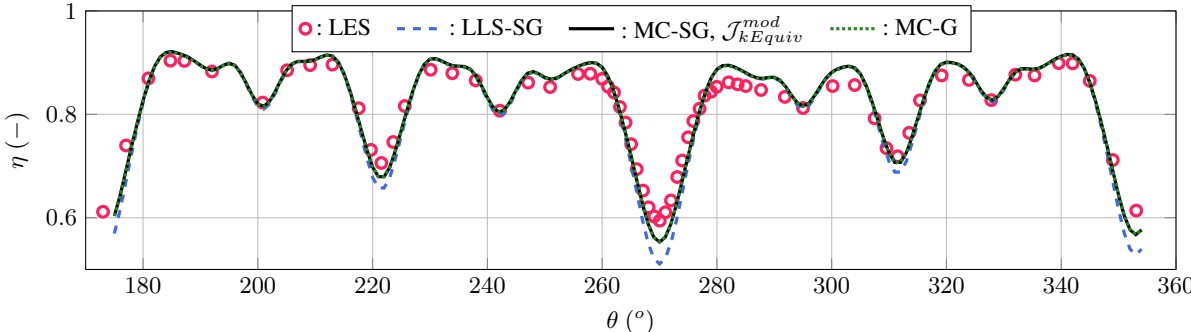

**Figure 4.** Comparison of the normalized Horns-Rev wind farm power output based on LES simulations from Porté-Agel et al. (2013), the LLS method together with the super-Gaussian model (LLS-SG), the momentum-conserving method together with the super-Gaussian model (MC-SG) and the momentum-conserving method together with the Gaussian model (MC-G)

that the poor results obtained in Lanzilao and Meyers (2022) for the same wind farm are mostly due to inaccuracies in the model calibration introduced in Cathelain et al. (2020).

For a more quantitative comparison, the RMSEs of the different analytical models against the LES results are provided in Table 3. The LLS method together with the super-Gaussian model has the highest level of error, around 3.7 per cent, while using the momentum-conserving approach, both with a Gaussian or a super-Gaussian model, the RMSEs falls below 2.5%. Differences between the two aforementioned models appear in the RMSEs only at the fourth decimal. Considering the large inter-turbine spacing in the Horns-Rev wind farm, this was expected, since both models use the same characteristic width, and the inter-turbine distances are large enough to have super-Gaussian orders very close to $k = 2$ at the rotor planes.

**Table 3.** Root-Mean-Square-Error of the analytical models against the LES results from Porté-Agel et al. (2013)

| MC-G | LLS-SG | MC-SG, $\mathcal{J}^{mod}_{kEquiv}$ |
|------|--------|------|
| 0.024 | 0.037 | 0.024 |

## 4 Conclusions

In this work, the momentum-conserving wake superposition method proposed in Bastankhah et al. (2021) was extended to super-Gaussian-type of velocity deficit models. An integral could not be resolved analytically, and an approximation has been proposed. This approximation is closer to numerical evaluations of the integral that the Gaussian assumption used in Bay et al.

(2022). Comparisons against large-eddy simulations of wind farms show a satisfactory agreement, allowing the simulation of large wind farms using the super-Gaussian wake model. Further studies will include an extensive validation of the resulting wind farm flow model, including closely-packed wind farms.

*Code and data availability.* The numerical results based on the analytical models can be made available on demand.

*Author contributions.* Frédéric Blondel derived the analytical solution introduced herein, performed the simulations and wrote the manuscript

*Competing interests.* The author declares no competing interests.

*Acknowledgements.* The author is grateful to Majid Bastankhah for the helpful discussions.

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
