# Peer review of "Brief communication: A momentum-conserving superposition method applied to the super-Gaussian wind turbine wake model"

_Wind Energy Science, 2022_

## Author Comment (AC1)

Dear reviewers,

First, I could like to thank you for revising this manuscript, and for your constructive remarks. The original paper has been modified accordingly.

Please find below a list of answers (in italic) to your comments and suggestions.

Best regards,

F. Blondel

Reviewer 1:

The paper extends the recent wake superposition model proposed in Bastankhah et al. 2021 by using a super-Gaussian distribution to express wake velocity deficit profiles, instead of a Gaussian distribution. The former is expected to improve results in the near-wake region. As discussed in the manuscript, the integral form of the conservation of momentum deficit for a waked turbine cannot be solved analytically for a super-Gaussian profile. However, the author used a numerical method as well as two approximate analytical methods to compute the integral. Results are compared with the LES data for two wind farm layout configurations. Overall, the paper is well-written with clear figures and interesting and critical discussion of results. So I recommend the publication provided that my below comments (in no particular order of importance) are addressed:

- Equation 6: I think "n" should be replaced with "k". Please check!

      → *Indeed there was a typo, that has been fixed*

- Equation 9: It is not clear how this equation is obtained. Please provide more clarification about how model coefficients (shown in Equation 9 and Table 1) are tuned.

      → *It was first decided not to provide too many details but to refer to the proceeding of Cathelain et al. However, it seems to make the present paper unclear and/or incomplete. More details have been integrated in the revised version (see lines 78→94)*

- Line 93: "… are estimated using a local rotor element area ponderated average". Please clarify what this means and how it affects obtained results.

      → *Clarifications have been integrated into the revised version, see lines 104→110*

Line 100: "... lead to superimposed velocity deficit". I am not sure if I understood what this sentence means. Please paraphrase it and elaborate what you mean here.

→ *The wording was indeed unclear. A new formulation has been integrated, see lines 120 and 121.*

Figure 2: It is shown that MC methods underpredicts the velocity deficit for the second turbine. Line 107 states that this is a consequence of the application of the "modified" momentum conservation. So please add the results for the "original" momentum conservation as I think this would strengthen your argument and also would be interesting for the reader to see how they are compared with each other.

→ *As mentioned in the text, the "original" formulation leads to very high velocity deficits, and potential unrepresentable numbers. Following the reviewer's advice, results based on the so-called "original" method have been included, see Figure 3. In order not to overload the original figure that already contained five different sources, a new one has been included, showing results from both "original" and "modified" formulation applied to the Gaussian model, the LES reference results, and also the super-Gaussian "modified" formulation.*

- The main contribution of this work is the inclusion of super-Gaussian profile. To more clearly show how significant this affects the results, please include original model predictions (using Gaussian profile) in both figures 2 and 3.

→ *It does indeed make sense to add these comparisons. Figures 3 and 4 now include these comparisons (there is a new figure 2, thus figures 2 and 3 from the preprint are now 3 and 4).*

Reviewer 2:

The proposed manuscript extends previous superposition-based wake models to super-Gaussian profiles. While such approach results in an integral with no analytical solution, the author proposes different approximations (numerical and analytical) that are evaluated against the LES results from Bastankhah et al. (2021). It is found that both approximations have a satisfactory agreement with the numerical data.

I consider the proposed model of interest for the community and the results sound. In consequence, I think the paper is suitable for publication on WES. Nevertheless, I have several remarks that should be addressed prior to publication:

- As also proposed by another anonymous reviewer, figure 3 should also show the Gaussian and numerical models. This would support the conclusions drawn by the author.

  → *As answered to reviewer 1, these comparisons have been included*

- Also, both in figures 2 and 3 the agreement between different models and the LES should be quantified to show the overall performance of them.

  → *For both cases, tables including the RMSEs have been included. This indeed supports the conclusions drawn in the preprint.*

- Which is the influence of the rotor disk discretization (line 93) in the results? The author states a 12x12 polar grid has been used but this number is not discussed.

  → *The way rotor disks are discretized in such wind farm flow solver indeed has an impact on the results. This point is usually not discussed in publications, and I felt it was interesting to include it. However, presenting a convergence study based on the rotor disk discretization is out of the scope of the present study (but will be included in future work). The choice of the 12x12 is a bit conservative: lower discretisations (6x6, typically) already lead to almost converged results. Nevertheless, a short discussion about polar discretization and its impact on the results has been included in the revised paper, see lines 103 to 110.*

- I am not convinced about the relevance of figure 1. It is not expected that an approximated super-Gaussian profile will perform better than a Gaussian one? Also, the maximal relative error for the kEquiv model still seems to be high. I propose the author gives such value in the main text. Furthermore, on the same topic, which are the values of kEq used in figures 2 and 3?

  → *There seems to be a small confusion here regarding the value of kEq used in the simulation. As stated in the text line 62, kEq is the result of the average of the Gaussian order from turbine "l" and turbine "n". Thus, it depends on the distance between the considered point and turbines "l" and "n", and of course thrust coefficient and mean turbulence intensities at rotor "l" and "n" planes. In figure 1, "Gauss" refers to the integral approximation $J_{Gauss}^{mod}$, in opposition to $J_{Gauss}^{kEquiv}$. It shows the error committed on the integral approximation. In both cases, no velocity deficit nor velocity profiles are used. It is really about quantifying the error on this specific integral term, for which we could not find an analytical solution. In this regard, I think it makes sense to keep this error quantification in the paper.*

- I find section 3 too short. For instance, the parameter a_f is not defined and its relevance never explained. Also, can the author be more specific about the poor performance previously exhibited by the super-Gaussian model?

  → *Reviewer 1 also commented on this part. A short discussion on the poor performance of the model in previous studies has been included in the text (see lines 78 to 82). This discussion also highlights the differences between the procedure employed in Cathelain et al. and the present: enforcing k=2 as a*

*boundary condition in the far wake. A more complete description of the model is also provided: it should be self-consistent, i.e., one should be able to implement the model from the information provided in the text, lines 82 to 91.*

- This is just a suggestion, but it would help the reader to add a diagram, maybe as an inset in a figure, where the layout of the farm and the parameters n and i are shown.

  → *This is indeed a good suggestion, and a diagram has been included (see Figure 2 in the revised version).*

- The caption of figure 2 should give more details. For instance, where are the turbines placed? Also, before the streamwise distance $5x/d_0$ all models collapse? And why the velocity deficit increases after $x/d_0=2$?

  → *Figure 3 caption (originally figure 2) has been extended, and, together with the new schema of the farm (figure 2 of the revised paper), it should be more clear now. The results are discussed more extensively, see lines 116 to 121.*

- The manuscript has some typos and problems with definitions. In line 43 should say 'interestingly' instead of interesting. In line 45 and equation 4 the velocity U is not defined and shown alternatively with and without capital letters. In line 65 it is not clear if the characteristic widths are or not normalized.

  → *These comments have all been taken into account (except "interesting", which I could not find in the original text). Complete proofreading of the manuscript has been performed.*

---

## Author Response (AR2)

Dear editors,

The requested changes have been made. Thank you !

Frédéric Blondel